# Microstructure Evolution and Dynamic Recrystallization Behavior of Mg-Gd-Y-Zn-Zr Alloy during Rotating Backward Extrusion

**DOI:** 10.3390/ma15031057

**Published:** 2022-01-29

**Authors:** Yali Duan, Jianmin Yu, Beibei Dong, Huifang Zhang, Mu Meng, Mei Cheng, Zhimin Zhang, Baohong Zhang, Hongyuan Hao, Ping Xu, Huiling Liu

**Affiliations:** 1College of Materials Science and Engineering, North University of China, 3 Xueyuan Road, Taiyuan 030051, China; yay20001999@163.com (Y.D.); zhf990715@163.com (H.Z.); 15203402460@163.com (M.M.); chengmei@nuc.edu.cn (M.C.); ZZMNUC@163.com (Z.Z.); zhangbh@nuc.edu.cn (B.Z.); Haohongyuan990725@163.com (H.H.); 2Department of Mechanical and Electrical Engineering, Taiyuan City Vocational College, 3 Xinghua Street, Taiyuan 030027, China; 3Norinco Group Air Ammunition Research Institute Co., Ltd., Harbin 150030, China; pingxu907@163.com; 4Mechanics Institute, Jinzhong University, Jinzhong 030619, China; liuhuiling@jzxy.edu.cn

**Keywords:** magnesium alloys, gradient structure, texture evolution, rotation backward extrusion (RBE), dynamic recrystallization (DRX)

## Abstract

Rotating backward extrusion (RBE) is one of severe plastic deformation (SPD) methods used to produce cylindrical components with a very large strain by a single pass. In this study, the microstructure and texture evolution in the different regions of Mg-12Gd-4Y-2Zn-0.5Zr (wt.%) alloys via RBE process were investigated by using optical microscopy (OM), X-ray diffraction (XRD), scanning electron microscopy (SEM) and electron back-scatter diffraction (EBSD). The results showed that the cup-shaped sample formed by RBE process exhibited typical gradient microstructure expanding from its inner wall to outer wall along the radial direction (RD). The average grain size of the RBEed sample decreased when the radius decreased from the edge region to the center region along the RD, which was attributed to the different strains and strain rates in the different regions. It also could be observed that the center region showed highest deformation and the edge region exhibited the lowest deformation in the RBEed sample along the RD. In addition, the grain refinement mechanisms of the experimental alloy containing long-period stacking ordered (LPSO) phases after RBE with 100 N were continuous dynamic recrystallization (CDRX), discontinuous dynamic recrystallization (DDRX) and particle stimulated nucleation (PSN).

## 1. Introduction

As the lightest structural and engineering material, magnesium (Mg) and Mg alloys have attracted considerable attention in the aeronautics and astronautics industries [1,2,3,4]. However, applications of Mg alloys are still limited owing to their low strength and poor ductility both at room temperature (RT) and high temperature (HT) because of their special hexagonal close-packed (HCP) structure [5,6,7,8]. Therefore, in order to improve their tensile strength with excellent ductility simultaneously [9] and make full use of their benefits, the hot extrusion method is widely applied in the manufacturing procedure for Mg alloys. Adding rare-earth (RE) elements such as Gd [10], Y [11] and Nd [12] to pure Mg can also effectively enhance its mechanical properties. In addition, introducing Zn element to the Mg-Gd based alloys can generate strengthening phases and long-period stacking ordered (LPSO) phases [13]. An ultra-high yield strength of 417 MPa and moderate ductility of 12.9% were obtained in Mg-8.2Gd-3.8Y-1.0Zn-0.4Zr alloy through a method of decreased-temperature MDF processing followed by ageing treatment by Tong et al. [14]. The high tensile strength was attributed to the grain refinement, dynamic recrystallization and broken secondary phase particles. Zhang et al. [15] developed Mg-3.5Sm-2Yb-0.6Zn-0.4Zr alloy containing low RE with high strength using an extrusion and ageing treatment. The peak-aged sample exhibited superior tensile yield strength (TYS) of 449 MPa and acceptable failure elongation (FE) of 4.9%.

Grain refining and texture weakening are valid methods to enhance the mechanical properties and improve the forming-ability of Mg alloys [16]. Additionally, the severe plastic deformation (SPD) methods including multi-directional forging (MDF) [17,18], repetitive upsetting-extrusion (RUE) [19], equal channel angular pressing (ECAP) [20,21], high pressure torsion (HPT) [22,23], accumulative roll bonding (ARB) [24,25], etc., are regarded as effect techniques to obtain ultra-fine grain (UFG) via accumulating strain. However, there are disadvantages of high production cost and low production efficiency during the SPD process. Thus, in order to overcome these shortcomings, a new SPD method named rotation backward extrusion (RBE) is proposed, which can obtain large strain through a single-pass deformation. The RBE technique is a combination of conventional backward extrusion (CBE) and torsion deformation. It is reported that the activation energy when introducing shear stress in the Mg alloy under the torsional deformation process is similar to the self-diffusion activation energy of Mg lattice during the thermal process, and the deformation mechanism is attributed to the diffusion behavior [26]. In addition, during the torsion deformation process, the critical strain is slightly smaller than that during compression deformation.

As we all know, the simple extrusion process is based on the increase in the extrusion ratio to obtain a large strain. However, there is a limitation during the extrusion process, because the larger the extrusion, the more massive the equipment that is required. Therefore, some extrusion methods with the introduction of torsional deformation have attracted the attention of more and more scholars [27,28,29,30,31]. The working principle of rotating extrusion deformation is that a die is rotated during the hot extrusion process in order to reduce the forming load and accumulate strain at the same time. In addition, compared to the traditional extrusion technique, the same strain could be achieved by single pass via the rotating extrusion method. Dong et al. [32] reported the microstructure and the texture evolution of AZ80 alloy cylindrical tube by RBE with different rotating revolutions, and found that the average grain size was decreased and the proportion of dynamic recrystallization (DRX) was increased gradually with the increase in rotating revolutions. Che et al. [16] studied the effect of processing parameters of a cup-shaped AZ80 Mg alloy via the RBE method; the microstructure and texture evolution were investigated simultaneously. It could be concluded that the average grain size was decreased gradually as the rotating revolutions increased or temperature decreased. 

Under certain torsion deformation conditions, the strain rate in any annular region can be maintained constant. It implies that the strain exhibits a gradient microstructure from the center to the edge region of the sample. However, the gradient features of RBEed Mg alloys containing LPSO phases have not been fully investigated. Thus, the primary purpose of this paper was focused on the microstructure evolution and DRX behavior of the Mg-12Gd-4Y-2Zn-0.5Zr (wt.%) alloys via the RBE process.

## 2. Materials and Methods

The composition of Mg-12Gd-4Y-2Zn-0.5Zr alloy exhibited in Table 1 was defined at 760 °C in an argon atmosphere, and the cast rod was obtained by employing a semi-continuous casting process. Then, the cast rod with the size of Φ 330 mm × 1000 mm was homogenized at 520 °C for 16 h. Finally, the experimental samples with the size of Φ 21 mm × 24 mm were taken out from the as-cast ingot. Figure 1a shows the principle of the RBE technique: put initial cylindrical blank into the die and the backward extruded behavior between the punch and the die will occur. Meanwhile, the die was continuously rotated by wising the motor drive during the extrusion process. Figure 1b shows the shape of the final sample obtained via the RBE process.

The deformation temperature was 450 °C and the heating rate was 1.5 °C/s during the RBE process. Once the desired temperature was reached, the sample was left resting for 5 min, and then the RBE test was started. The experimental parameters were that the axial extrusion speed of the punch was 6 × 10^−2^ mm/s, the stroke was 20 mm, the die speed was 209.4 × 10^−2^ rad/s (100 turns) and the total RBE deformation time was 5 min. The RBE processing was conducted on a Gleeble-3500 torsion system. The microstructure observation was along the radial direction (RD) of the RBEed samples via optical microscope (OM, Zeiss Axio Imager-A2m, Germany) and scanning electron microscope (SEM, Hitachi-SU5000, Japan) equipped with EDAX-TEAM electron back-scatter diffraction (EBSD) system, as shown by red dots in Figure 1. In addition, in order to obtain high-quality EBSD data, the observation planes were metallographically prepared by using SiC papers and 50 nm diameter silica suspension. Subsequently, the residual stress layers were polished by employing an ion polishing machine (Leica EM Res 102, Germany) at 6.5 kV and 3 mA. The scanned areas of the RBEed samples were evaluated via EDAX-OIM analysis software to determine the crystallographic texture, grain size, misorientation angle, etc. In addition, grain sizes of less than 10 μm were regarded as DRXed grains in this paper.

## 3. Results

### 3.1. Microstructure Observations of the Homogenized Alloy

The OM and SEM images of the Mg-12Gd-4Y-2Zn-0.5Zr alloy after homogenization treatment are shown in Figure 2a,b, respectively. The average grain size of the experimental alloy was as 80 μm. The homogenized alloy mainly consisted of Mg matrix, Mg_5_RE phases, lamellar LPSO phases, block-shaped LPSO phases and cubic RE-enriched phases. It was obvious that the lamellar LPSO phases with the same orientation were inside the grains, the block-shaped LPSO phases were distributed along the grain boundaries, and the cubic RE-enriched phases were distributed in the Mg matrix randomly. Figure 2c exhibits the X-ray diffraction (XRD) pattern of the initial alloy and the RBEed sample after 100 N. The results show that no new precipitation phases were found in the RBEed sample after 100 N compared with the initial alloy.

### 3.2. Microstructure Evolution during the RBE Process

The strain gradually increases and the morphology of the metallographic structure varies from the inner circle to the outer circle during the RBE process. Figure 3a displays the microstructure of the cross-section in the different zones along the radial direction (RD), and Figure 3b exhibits the microstructure of the vertical section in the different zone profiles along the extrusion direction (ED) of the RBEed specimen after 100 N (N represents number of revolutions after the RBE process), respectively. The RBEed sample could be divided into six zones along the RD, and the zones were named region 1, region 2, …, region 6 according to the different distances from the center, as shown in Figure 1a. The microstructure from the inside to the outside of the sample itself was embedded during the entire RBE process, and the RBEed sample exhibited a gradient structure from the center to the region for a certain section, which was attributed to the variational shear stress during the RBE process. In addition, it is obvious that the deformation streamline covered almost entirely region 1 and region 2. Region 3 was in the transition zone, and several grains exhibited an elongated microstructure along the RD. Additionally, almost no deformation behavior appeared in regions 4, 5 and 6.

Figure 4 shows the highly magnified microstructure of the vertical section in the different zone along the RD. The regions 1, 2, 3, 4, 5 and 6 were approximately 0.9 mm, 1.8 mm, 2.7 mm, 3.6 mm, 4.5 mm and 5.4 mm away from the center to the edge of the RBEed sample with 100 N. It is obvious that the grains were refined and exhibited a significant gradient microstructure, with the largest strain observed at the center and the lowest deformation of the RBEed sample along the RD. The average grain size of the RBEed sample was 3.0 μm, 9.4 μm, 22.8 μm, 48.1 μm, 55.7 μm, and 56.0 μm in region 1, region 2, region 3, region 4, region 5 and region 6, respectively. In addition, the block-shaped LPSO phases were gradually broken into small fragments from the edge to the center due to the mechanical crushing and shear deformation. The microstructure of the RBEed sample in region 1, region 2, region 3 and region 4 consisted of a large number of coarse undeformed grains and a small number of DRXed grains. The morphologies of the LPSO phases were both lamellar-shaped and block-shaped in regions 3, 4, 5 and 6. The fine DRXed grains were mainly found around the block-shaped LPSO phases; this phenomenon could be regarded as the particle stimulated nucleation (PSN) mechanism, and the block-shaped LPSO phases provided the nucleation position for the DRXed grains. Region 6 (5.4 mm) and region 5 (4.5 mm) showed almost no deformed grains, and only a few fine DRXed grains could be observed along the original grain boundaries. The fine second particles of the Mg5RE phases were distributed along the grain boundaries. The average grain size in the region 5 was slightly smaller than that in region 6, and the number fraction of the DRXed grains in region 5 and region 6 were 12.7% and 11.3%, respectively. In region 4 (3.6 mm), the degree of DRX was more pronounced compared to that in regions 5 and 6, and part of the second phases in the grain boundaries began to fragment. In region 3 (2.7 mm), the number fraction of the DRXed grains further increased, and the large-sized lamellar LPSO phases started to divide into several small-sized lamellar LPSO phases due to the occurrence of the DRXed grains and the kink behavior. Subsequently, region 3 could be regarded as a transition zone, and it transmitted the deformation from the center to the edge during the RBE process. In region 2 (1.8 mm), most of the grains were refined and exhibited typical bimodal microstructure that consisting of coarse undeformed grains and fine DRXed grains. In region 1 (0.9 mm), it is obvious that all the grains were refined and the second phases were completely fragmented at the time. The LPSO phases only existed in regions 1 and 2 with a block-shaped morphology, and the microstructure was uniform. However, the streamlines along the RD were not continuous. The number fraction of the DRXed grains was 100% and 72.1 % in regions 1 and 2, respectively. Furthermore, it could be concluded that DRX occurred completely in region 1.

Figure 5 shows the EBSD inverse pole figure (IPF) maps in the cross-section of the different zones along the RD. The different colors of the grains manifested different orientations, and the LPSO phases were shown by a large black area. The low-angle grain boundaries (LAGBs, 2° ≤ θ < 15°) and the high-angle grain boundaries (HAGBs, θ > 15°) were characterized by white lines and black lines, respectively, where θ represents the misorientation angle between adjacent crystal grains. The EBSD results revealed that the RBEed sample in region 1 consisted of fine equiaxed grains, which were attributed to the well-developed DRX. It could be observed that the complete DRX behavior had been achieved, and the whole grains in region 1 could be regarded as fine DRXed grains. A typical bimodal microstructure was shown in regions 2, 3, and 4, and the coarse undeformed grains were surrounded by the fine DRXed grains. Only a small amount of DRXed grains could be found in regions 5 and 6. Wang et al. [32] found that the microstructures were inhomogeneous on the cross-section of the sample; in addition, the microstructures also exhibited significant gradient distribution due to the uniform shear strain [33]. It could be concluded that the generation of the gradient microstructure was mainly attributable to the different shear strains and strain rates in the different regions, and the parameters such as strain and strain rate exhibited the same tendency. The specific performance was that the above two parameters decreased from the center region 1 to the edge region 6 along the RD. In the RBE process, the strain and strain rate decreased with increases in radius, and the streamlines also weakened with the radius increases and gradually became discontinuous at the same time. In addition, it was obvious that not only the coarse undeformed grain boundaries but also the fine DRXed grain boundaries exhibited serration and began to bulge in different regions of the RBEed sample with 100 N, which was consistent with the characteristics of the discontinuous dynamic recrystallization (DDRX) mechanism [32,34,35]. Therefore, it could be concluded that the DDRX behavior occurred during the RBE process. Moreover, the area fraction of the LAGBs and HAGBs were 43.6% and 56.4%, 39.3% and 62.7%, 46.1% and 53.9%, 59.0% and 41.0%, 63.0% and 37.0%, and 56.8% and 43.2% in regions 1, 2, 3, 4, 5, and 6, respectively. The LAGBs resulted from the accumulation of dislocations and/or the formation of sub-grain boundaries in the deformed microstructure. With the strain increases, the LAGBs transformed into HAGBs by absorbing the mobile dislocations, eventually modifying the sub-grains into new DRXed grains [36,37]. It was also found that continuous dynamic recrystallization (CDRX) occurred in the RBE process. The reason for the generation of the refined grains was the joint action of DDRX and CDRX.

Figure 6 displays the corresponding grain size distribution and (0001) and (10-10) pole figures (PF) in the cross-section of the different zones along RD. It is obvious that all the regions of the RBEed sample with 100 N exhibited random texture. The maximum texture intensity of the RBEed sample along the RD was 2.649, 2.662, 4.973, 8.917, 10.086 and 10.813 in region 1, region 2, region 3, region 4, region 5 and region 6, respectively. It could be concluded that the maximum texture intensity of the RBEed sample was gradually weakened with the radius decreases, which was attributed to the higher number fraction of the DRXed grains. In addition, the RBEed sample in region 1 exhibited two main components: one was a relatively strong texture component at [10−10] and the other, a weak component at [2−1−10]. The maximum texture intensity of the RBEed sample in region 2 showed strong components between [10−10] and [2−1−10]. The RBEed sample in region 3 displayed two main components: one was a weak component between [0001] and [10−10], and the other was a strong component between [0001] and [2−1−10]. The RBEed sample in region 4 revealed a maximum texture component distributed between [0001] and [2−1−10]. Additionally, the RBEed sample in region 5 exhibited three main components: one of them was the peak texture component distributed along [10−10], while the others were relatively weak components at [0001] and [2−1−10].

### 3.3. Dynamic Recrystallization (DRX) Mechanism during the RBE Process

It is reported that the DRX mechanism can be strain-induced nucleation [38], particle-stimulated nucleation [39], twining-induced recrystallization [40], and shear-band nucleation [41]. In order to further elucidate the DRX process on the deformation texture, the zone of region 5 was selected to perform a detailed analysis, as shown in Figure 7a. The (0001) PF and IPF exhibited a typical bimodal microstructure (Figure 7b,c), which consisted of coarse unDRXed grains and relatively fine DRXed grains. The number fraction of the unDRXed grains and DRXed grains was 88.7% and 11.3% in region 5, respectively, which indicated an inhomogeneous grain distribution. In addition, it was also found that a large number of sub-grain boundaries existed inside the grains. To better display the texture characteristics, the crystallographic orientations of the unDRXed grains and DRXed grains are highlighted by blue color and red color, respectively. The RBEed sample in region 5 was characterized by a strong [0001] texture component and [10−10] fiber component; however, the [2−1−10] component was relatively weak. It was significant to find that the unDRXed grains exhibited a strong texture with [0001] fiber orientation, whereas the DRXed grains shows the spread and relatively random distribution. This confirms the results mentioned before, which indicated that the generation of the [2−1−10] texture components is related to the occurrence of the DRX behavior. The KAM map shown in Figure 7f reveals that region 5 with the high KAM value accumulated the generation of the sub-grain boundaries, which provided a strain gradient sufficient to trigger the CDRX mechanism. The region 5 with high-stress concentration was located in the neighborhood of the sub-grain boundaries, and the stress concentration was attributed to the interaction between the lamellar LPSO phases and the basal slips [42].

In addition, in order to shed some light on the formation of the DRXed grains, as well as on their preferred orientation, a typical unDRXed grain of G1 was further analyzed as shown in Figure 8. The line profile of the point-to-origin arrow AB shows that the misorientation angle increased up to 11° (Figure 8b), which indicates the occurrence of the continuously changing orientation in G1 as shown from the 3D hexagons along the AB in Figure 8a. Moreover, Figure 8c presents the KAM map of the grain of G1. The results reveal that the local stress concentration were located along the sub-grain boundaries, which could be attributed to the interaction of dislocation entanglements between the dislocations and the LPSO phases. It also offered sufficient strain energy to trigger the CDRX process.

Moreover, the researches mentioned that the LPSO phases exhibited a strong plastic anisotropy because the deformation modes were limited [43,44]. Yu et al. [45] found that the generation of the strong stress concentration was attributable to the incompatibility of deformation, and the phenomenon of stress concentration mainly occurred at the boundaries between the Mg matrix and LPSO phases during the deformation process. The strong stress concentration could lead to the generation of fine DRXed grains and a relatively weakened texture intensity. Thus, the degree of stress concentration was higher in region 1 than in region 6. The grain of G2 with low content LPSO phases and the grain of G3 with high content LPSO phases shown in Figure 9 were chosen for further analysis, as shown in Figure 10 and Figure 11. The line profile of the point-to-origin along the black arrow in Figure 10b shows that the misorientation angle increases gradually, up to about 8°. It indicates that the continuous change in orientation takes place in G2 and suggests that the dislocations occur in the grain with high activity, as shown by the 3D hexagons along the AB in Figure 10a. Figure 11c exhibits the IPF of G2, and it can be found in the grain clusters at [0001]//ED.

Compared with the grain of G2, the grain of G3 containing higher content of LPSO phases showed a preferred selection for the non-basal orientation shown in Figure 11. The misorientation angle of G3 was lower than that of G2, indicating the dislocations are easily accumulated and the DDRX behavior is easier to generate.

## 4. Conclusions

In this work, the deformation behavior of Mg-12Gd-4Y-2Zn-0.4Zr alloy via RBE with 100 N was investigated. Furthermore, the microstructure and texture evolution of the cup-shaped sample were also studied. The effects on the grain refinement and the DRX mechanism of the alloy during the RBE process were described. The conclusions can be summarized as follows:

(1) The cup-shaped sample by RBE process exhibited typical gradient structure along the RD from the inner wall to the outer wall. The average grain size decreased from 56.0 μm in region 6 to 3.0 μm in region 1.

(2) The number fraction of the DRXed grains increased significantly with the decrease in radius from region 6 to region 1. The grains in region 1 were all DRXed grains, and it could be regarded that the complete DRX process occurred in region 1.

(3) The maximum texture intensity of (0001) basal plane was gradually weakened from the outside region 6 towards the inside region 1 along the RD of the cup-shaped sample. In particular, the (0001) basal texture in region 5 transitioned to the [10−10] and [2−1−10] double fiber orientation in region 2, and then to the strong [10−10] texture component in region 1.

(4) The grain refinement process of the experimental alloy was investigated. Initially, the DRX behavior occurred in the grain boundaries of several grains; then, the dislocations started occurring in the grains of lower content LPSO phases and the larger-sized LPSO phases were decomposed into dispersed short fibers. The deformation of the dispersed LPSO phases interacted with the accumulated dislocations and formed sub-grain boundaries. The reason for the grain refinement was mainly attributed to CDRX, DDRX and PSN mechanisms.

## Figures and Tables

**Figure 1 materials-15-01057-f001:**
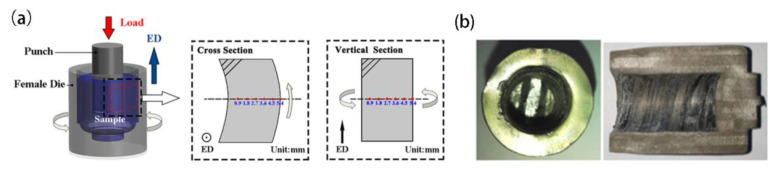
(**a**) The principle of the rotating backward extrusion (RBE) process, (**b**) the shape of the manufactured sample after the RBE process.

**Figure 2 materials-15-01057-f002:**
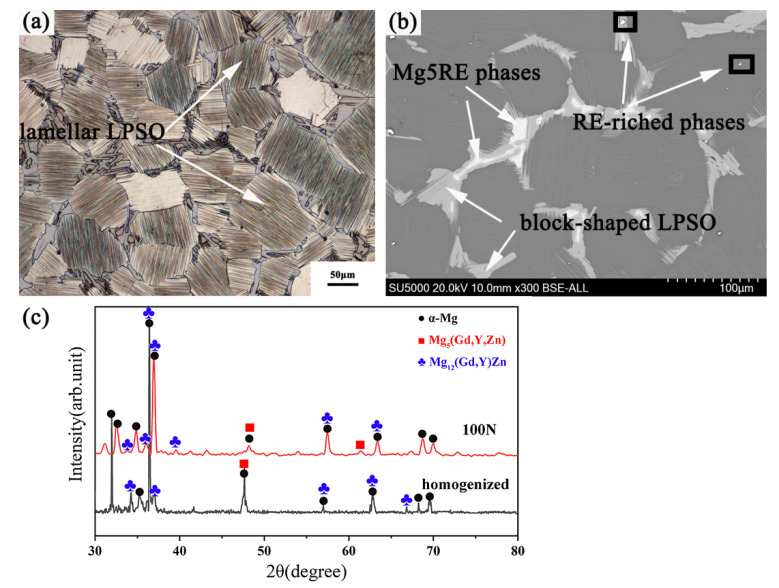
(**a**) The OM image of the as-homogenized Mg-12Gd-4Y-2Zn-0.5Zr alloy, (**b**) the SEM image of the Mg-12Gd-4Y-2Zn-0.5Zr alloy, (**c**) X-ray diffraction (XRD) pattern of the initial alloy and the RBEed sample after 100 N.

**Figure 3 materials-15-01057-f003:**
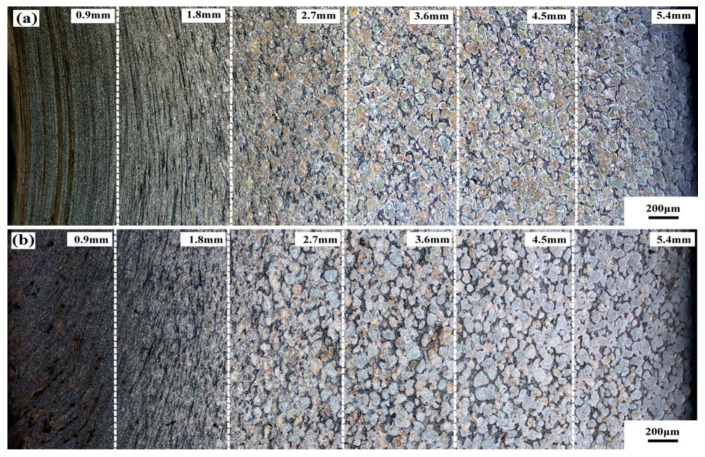
(**a**) The OM images of the cross-section in the different zones along the radial direction (RD), (**b**) the microstructure of the vertical section in the different zone profiles along the extrusion direction (ED) of the RBEed sample after 100 N.

**Figure 4 materials-15-01057-f004:**
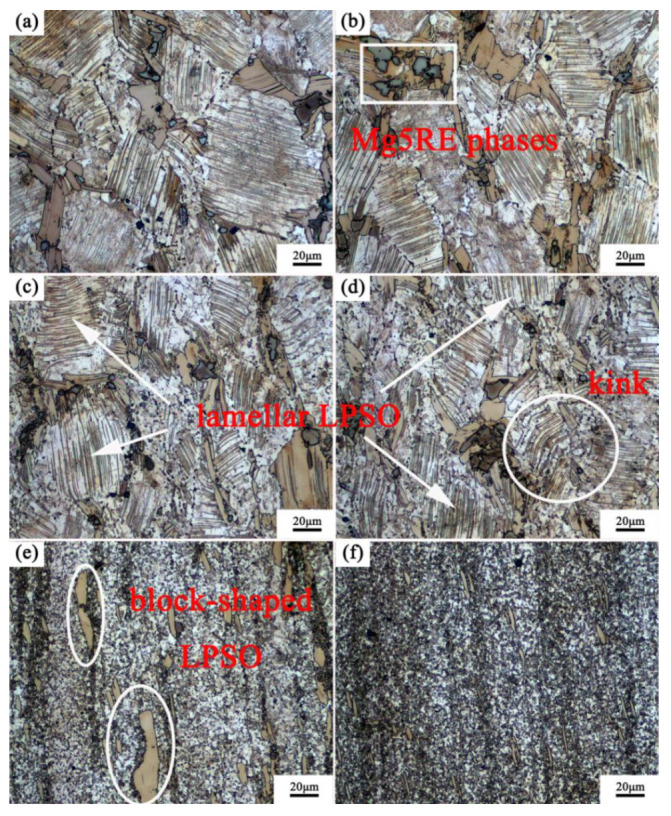
The OM images of the cross-section in the different zones along the RD, (**a**) region 6, (**b**) region 5, (**c**) region 4, (**d**) region 3, (**e**) region 2, (**f**) region 1.

**Figure 5 materials-15-01057-f005:**
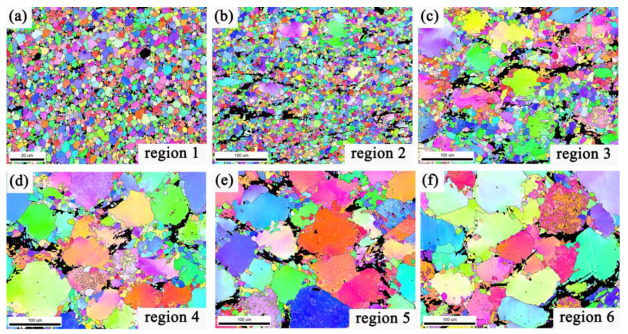
The EBSD maps of the cross-section in the different zones along RD, (**a**) region 1, (**b**) region 2, (**c**) region 3, (**d**) region 4, (**e**) region 5, (**f**) region 6.

**Figure 6 materials-15-01057-f006:**
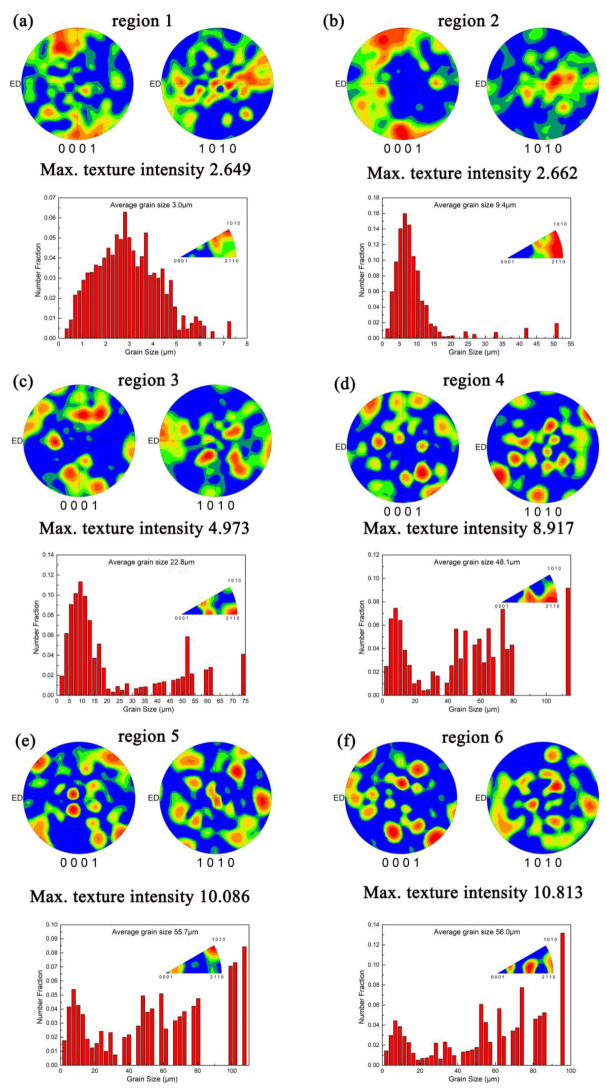
The corresponding grain size distribution and (0001) and (10-10) pole figures (PF) in the cross-section of the different zones along the RD.

**Figure 7 materials-15-01057-f007:**
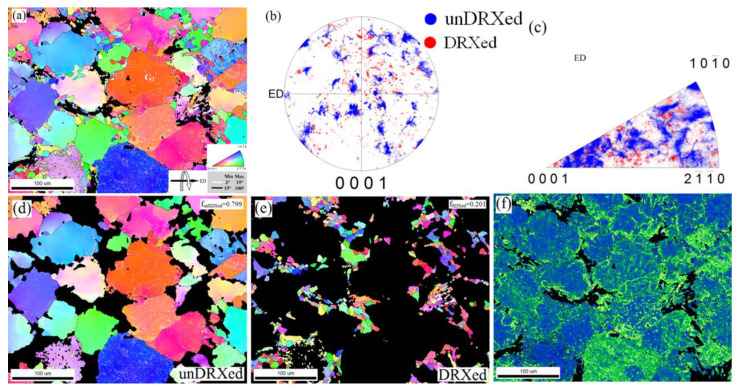
EBSD results of the RBEed sample in the region 5 containing DRXed grains and unDRXed grains, (**a**) IPF maps, (**b**) (0001) pole figure (PF), (**c**) inverse pole figure (IPF), separately highlighted IPF maps of (**d**) unDRXed regions and (**e**) DRXed regions, (**f**) KAM map.

**Figure 8 materials-15-01057-f008:**
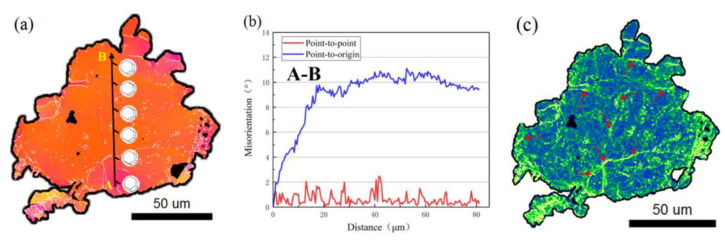
The unDRXed grain G1 in region 5 of the RBEed sample, (**a**) IPF map, (**b**) line profile of the orientation angle along the AB arrow depicted in (**a**), (**c**) KAM map of G1.

**Figure 9 materials-15-01057-f009:**
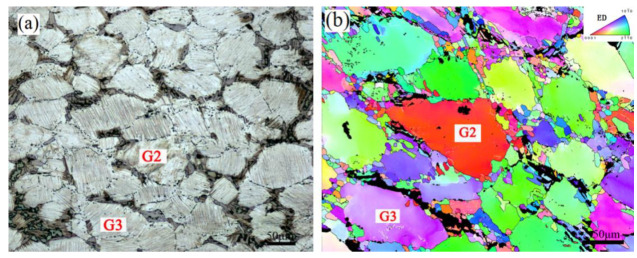
The (**a**) OM image and (**b**) IPF map of the grain with low content LPSO phases (G2) and high content LPSO phases (G3).

**Figure 10 materials-15-01057-f010:**
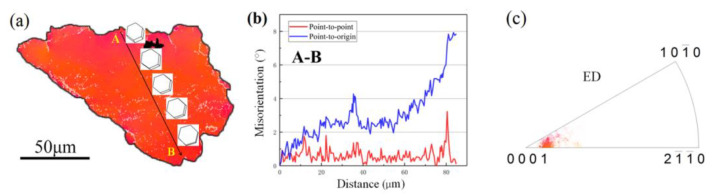
(**a**) Inverse pole figure (IPF) map of grain G2 containing low content LPSO phases, (**b**) line profile of the misorientation angle of the AB arrows depicted in (**a**), (**c**) IPF of the grain G2.

**Figure 11 materials-15-01057-f011:**
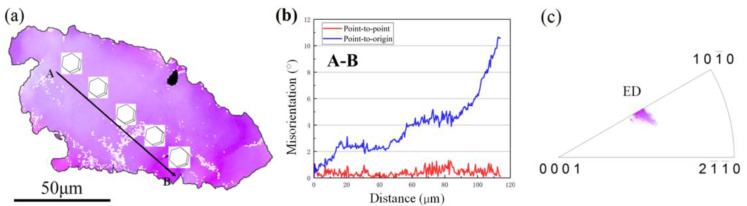
(**a**) Inverse pole figure (IPF) map of grain G3 containing high content LPSO phases, (**b**) line profile of the misorientation angle of the AB arrows depicted in (**a**), (**c**) IPF of grain G3.

**Table 1 materials-15-01057-t001:** The chemical composition of Mg-12Gd-4Y-2Zn-0.5Zr alloy.

Elements	Gd	Y	Zn	Zr	Si	Cu	Mg
wt.%	12.00	4.00	2.00	0.50	<0.01	<0.01	Bal.

## Data Availability

Not applicable.

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
