# Peer review of "Microstructure Evolution and Dynamic Recrystallization Behavior of Mg-Gd-Y-Zn-Zr Alloy during Rotating Backward Extrusion"

_materials, 2022, doi:10.3390/ma15031057_

Round 1

Reviewer 1 Report

Interesting article that needs a few minor revisions.

  1. Table 1 is missing from the article. How was the chemical composition of this alloy determined. Why at a temperature of 760oC?
  2. Figure 2 shows the X-ray diffraction (XRD) pattern. Only one peak corresponds to the Mg5RE phase regardless of the alloy state. However, most of the peaks correspond to the Mg12 (Gd, Y) Zn phase, which is not present in the photos of the structures and in the text. This requires some explanation
  3. The authors wrote “The average grain size of the RBEed sample is 3.0 μm, 9.4 μm, 22.8 μm, 48.1 162 μm, 55.7 μm, and 56.0 μm in the region1, region 2, region 3, region 4, region 5 and region 163 6, respectively.” What was the measurement error?
  4. It should be clarified what the authors mean by the term "the kink behavior"?
  5. The authors wrote„The grain of G2 with low content 295 LPSO phases and the grain of G3 with high content LPSO phases are chosen for further 296 analysis as shown in Fig. 11.” On what basis are they marked in Fig. 11a? There are no significant differences there.
  6. The article could use a table showing the differences in G1, G2 and G3 grains, along with showing them in the LM or SEM photos.

Author Response

Thanks for your suggestion, and we have corrected it.

Reviewer 2 Report

This is a well written article with well presented and carefully constructed and reported experiments and results.  The authors have demonstrated that torsion can assist extrusion processes for obtaining nano grains.

The only criticism I have is that the authors have only displayed the results as a cause and effect presentation.   There is little novelty except of the most obvious kind.

What I would have liked to see are more insights 

(1) a deeper discussion of LPSO structure and implications on dislocation movement. Similar discussion in a different field for example is given in Dey, G. K. (1997) Micropyretic synthesis of tough NiAl alloys Metallurgical and Materials Transactions B, 28 (5). pp. 905-918. ISSN 1073-5615

(2) More details on the energy employed per unit volume to generate Figure 3 (a) and (b). Such discussions will allow comparison of energy for the grain size which will make this article very attractive.

(3) Comparison with casting techniques to compare grain refinement where rotation and force are used e.g. Microstructure refinement with forced convection in Aluminum and Superalloys, e.g. Journal of Materials Science,1985, vol. 20, page 3535-3544.  I think it also makes reference to magnesium alloys, or for example in Moderate pressure solidification: Undercooling at moderate cooling rates, Acta Metallurgica,1989, volume 37, page1509-1519.

(3) Particularly following the article on undercooling, it would have been nice to see some discussions on recrystallization temperatures and their changes with the type of shear and residual strain.

Such discussions will allow comparison of energy for the grain size which will make this article very attractive.

Author Response

Dear editor and reviewer:

Thanks for your suggestion, the main contents of this paper are as follows:

Rotation backward extrusion is a new type of severe plastic deformation that combines high-pressure torsion and back-extrusion processes, which could effectively refine grains. It can be used to prepare components, breaking the current state that SPD is only used in the laboratory to make small-sized sheet samples.

In this work, we applied this new method to Mg-12Gd-4Y-2Zn-0.4Zr alloys. The microstructure evolution and dynamic recrystallization behavior of the cup-shaped RBEed sample were studied. In addition, the effect of LPSO phases on grain refinement and dynamic recrystallization (DRX) mechanism during the RBE process were further investigated.

As for your question, we are carrying out relevant research, and the specific content will be published in our subsequent articles.
